# Albumin-corrected anion gap as a predictive marker for mortality in critically Ill cirrhosis patients: an analysis based on the MIMIC-IV database

Ce Xu[1], Guangdong Wang[2], Min Zhang[1], Lihong Lv[1], Mengyuan Chen[1], Xingyi Yang[ID][1]*

**1** Department of Gastroenterology Disease, Xian Ju People's Hospital, Zhejiang Southeast Campus of Zhejiang Provincial People's Hospital, Affiliated Xian Ju's Hospital, Hangzhou Medical College. Xian Ju, Taizhou, Zhejiang, China, **2** Department of Respiratory and Critical Care Medicine, First Affiliated Hospital of Xi'an Jiao tong University, Xi'an, Shanxi, China

* 375424576@qq.com

## Abstract

### Background

The prognostic value of Albumin-Corrected Anion Gap (ACAG) has been extensively examined across a range of diseases; however, its relationship with short- and long-term survival in critically ill cirrhotic patients remains poorly understood. This study aims to investigate and elucidate the association between ACAG levels and mortality risk in this patient population.

### Methods

Initial analysis involved a univariate assessment of 30-day mortality outcomes, followed by stratification of patient data using X-tile software. Multivariate modeling was employed to identify independent risk factors for mortality. Survival outcomes associated with ACAG levels were analyzed using Kaplan-Meier (K-M) survival curves, while diagnostic accuracy was evaluated through Receiver Operating Characteristic (ROC) curve analysis. Additionally, Restricted Cubic Spline (RCS) regression was utilized to investigate potential non-linear relationships between ACAG levels and mortality risk. Subgroup analyses further validated the interactions between ACAG levels and mortality outcomes in the context of liver cirrhosis.

### Result

This study analyzed 2,826 participants, stratifying them into elevated (>20) and normal (≤20) ACAG groups based on X-tile-derived optimal cutoff. Elevated ACAG significantly correlated with higher mortality at 30, 90, 180, and 365 days (P < 0.05). Cox regression confirmed ACAG as an independent mortality predictor, supported by

**Data availability statement:** All relevant data are within the manuscript and its Supporting Information files.

**Funding:** The author(s) received no specific funding for this work.

**Competing interests:** The authors have declared that no competing interests exist.

consistent hazard ratios (P < 0.05). K-M analysis revealed worse survival in the elevated ACAG group (P < 0.05). ROC curves indicated fair predictive value for cirrhosis mortality, while RCS analysis showed a linear ACAG mortality relationship. Subgroup analyses revealed no significant interaction effects between ACAG and demographic or clinical variables.

## Conclusion

ACAG stands out as a reliable prognostic indicator, showing a meaningful link to mortality rates among critically ill patients with cirrhosis.

---

## 1. Introduction

Liver cirrhosis, a chronic and progressive liver disorder, is defined by hepatic fibrosis and structural abnormalities, leading to eventual liver dysfunction and failure [1]. Globally, its prevalence has risen steadily, particularly in developing regions where alcohol misuse and viral hepatitis infections are widespread [2]. Severe cirrhosis is frequently associated with life-threatening complications, including hepatic encephalopathy, ascites, gastrointestinal bleeding, and hepatorenal syndrome, all of which significantly elevate mortality risk [3]. Epidemiological studies report a five-year survival rate of approximately 50% for cirrhotic patients without liver transplantation, with critically ill individuals, particularly those presenting with multi-organ failure, experiencing even higher mortality rates [4]. End-stage liver disease not only profoundly diminishes patients' quality of life but also places a substantial economic strain on healthcare systems worldwide [5].

The anion gap (AG), calculated as the difference between unmeasured plasma anions and cations, is a pivotal diagnostic parameter for assessing acid-base homeostasis [6,7]. While the reference range for AG typically falls between 8 and 16 mmol/L, cirrhotic patients frequently exhibit alterations in this value due to liver dysfunction-induced electrolyte imbalances and metabolic acidosis [8,9]. In critically ill cirrhotic patients, the complexity of acid-base disturbances is often heightened by liver failure and systemic complications such as infections, leading to more pronounced AG abnormalities [10]. Notably, hypoalbuminemia, a common feature in cirrhosis, may compromise the reliability of conventional AG measurements, necessitating adjustments for albumin levels [11,12]. Consequently, the ACAG has emerged as a more precise diagnostic tool, offering enhanced insights into acid-base status and prognostic implications. ACAG has demonstrated significant prognostic value in various clinical conditions, including heart failure [13], acute myocardial infarction [14], and sepsis [15]. Despite these advancements, the association between ACAG and all-cause mortality in critically ill cirrhotic patients remains insufficiently investigated, warranting further research to elucidate its clinical significance in this population.

This study utilizes the MIMIC-IV database [16] to investigate the association between the ACAG and mortality in critically ill cirrhotic patients, aiming to establish a more reliable prognostic tool for clinical application. Through comprehensive

correlation analysis, we seek to elucidate the clinical significance of ACAG in this population and contribute to improved patient management strategies.

## 2. Materials and methods

### 2.1. Database introduction

This study uses data from the MIMIC-IV database (v3.1), a public resource developed by MIT's Laboratory of Computational Physiology. It includes detailed records of over 90,000 ICU patients at BIDMC (2012–2022), covering test results, prescriptions, vital signs, and hospitalization durations. The database has IRB approval from Beth Israel Deaconess Medical Center (2001P-001699/14). Patient confidentiality is maintained through de-identification (real data replaced with arbitrary numbers), eliminating the need for ethical approval and informed consent. The author accessed the database after completing the CITI Program (certificate #67058598). De-identified data can be found in the supplementary material "database".

### 2.2. Inclusion and exclusion criteria

The diagnosis of cirrhosis was established based on the International Classification of Diseases (ICD) codes (ICD-9: 571.2, 571.5, 571.6; ICD-10: K70.30, K70.31, K74.3, K74.60, K74.69). Our study included 6597 patients with cirrhosis requiring ICU admission. We excluded patients with: (1) non-first ICU admissions; (2) ICU stays shorter than 24 hours; or (3) unavailable AG or albumin (ALB) data within 24 hours of admission. The final analysis included 2826 patients. **Fig 1** illustrates the study's inclusion and exclusion process.

### 2.3. Data collection and monitoring

We extracted data on demographic details, initial clinical vital signs, laboratory results, comorbidities, and treatment outcomes from the MIMIC-IV database, focusing on the first 24 hours of ICU admission. We meticulously tracked vital signs, including blood pressure and heart rate. Furthermore, an extensive array of laboratory indicators was documented,

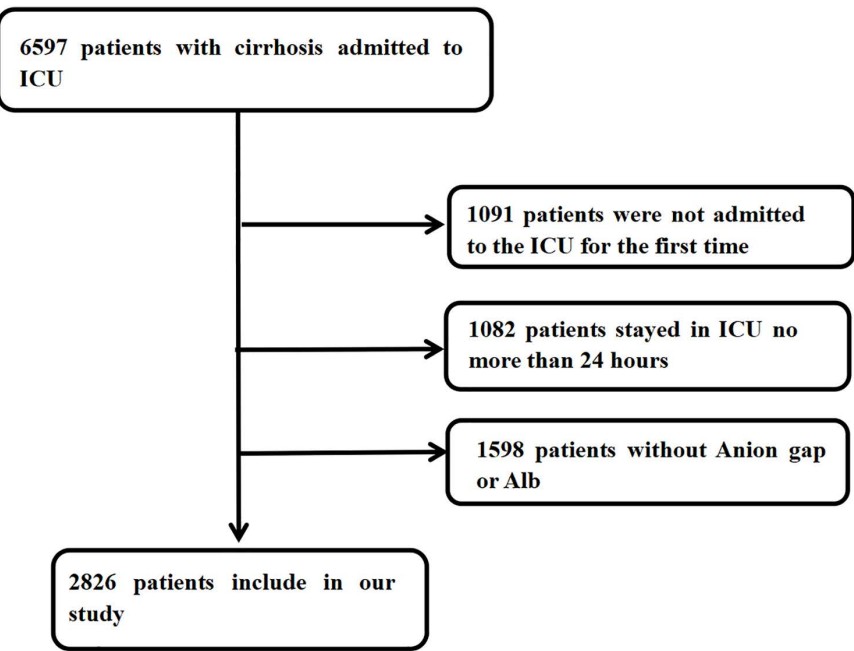

**Fig 1. A flow diagram of study participants.**

such as AG, albumin, total bilirubin, blood urea nitrogen (BUN), creatinine, blood glucose, white blood cell count (WBC), international normalized ratio (INR), transaminases, platelet count, sodium, potassium, and chloride levels. Data on complications closely associated with the prognosis of cirrhosis, which are of significant importance, were also collected. The complications associated with liver disease encompass ascites, variceal bleeding, hepatorenal syndrome (HRS), hepatic encephalopathy (HE), and spontaneous bacterial peritonitis (SBP). Additionally, we included established clinical scoring systems, namely the MELD [17] (Model for End-Stage Liver Disease) score as well as the SOFA [18] (Sequential Organ Failure Assessment) score, both of which are extensively utilized to evaluate disease severity and predict clinical outcomes. The ACAG served as the core variable in this study. The calculation formula for the ACAG (mmol/l) is: ACAG (mmol/l) = AG (mmol/l) + {4.4 − ALB (g/dl)} × 2.5. The model's continuous variable data were constructed using the average values of parameters recorded within the first 24 hours of ICU admission. Patients were followed up for a minimum of 365 days to evaluate both short-term and long-term prognostic outcomes. The primary endpoint of this study was all-cause mortality at 30-day, 90-day, 180-day, and 365-day intervals after admission.

## 2.4. Statistical analyses

The normality of continuous variables was evaluated using Shapiro-Wilk tests. Normally distributed data were presented as mean ± standard deviation (SD) and analyzed with independent samples t-tests, while non-normally distributed variables were expressed as median (interquartile range, IQR) and analyzed using Mann-Whitney U tests. Data with >20% missing values were excluded; those with ≤20% underwent random forest imputation. Random forest imputation is a powerful tool for data completion. Its non-parametric nature enables it to model complex nonlinear relationships in data without strict normality assumptions [19]. Also, compared to regression-based methods, it shows strong robustness to outliers, improving the accuracy of missing value estimation [20].

The cutoff value of ACAG was determined to be 20 mmol/L by optimizing 30-day post-admission mortality using X-tile analysis (v3.6.1, Yale University, USA) as well as referring to prior studies [21–23], with patients being divided into two groups: those with ACAG ≤20 mmol/L and those with ACAG >20 mmol/L. The Cox model was used for univariate and multivariate analyses to identify independent prognostic factors for 30-day mortality in cirrhosis patients. Results were reported as Hazard ratios (HRs) with 95% confidence intervals (CIs). KM curves were plotted, and the log-rank test compared the groups. ROC analysis evaluated the predictive accuracy of ACAG, AG, ALB, SOFA score, and MELD score for mortality. Sensitivity, specificity, and AUC were calculated. Restrictive cubic spline (RCS) analysis explored the relationship between ACAG and cirrhosis prognosis. Subgroup analyses assessed the impact of factors like age, sex, race, sepsis, AKI (Acute Kidney Injury), variceal bleeding, SBP, hepatic encephalopathy, HRS, and CRRT (Continuous Renal Replacement Therapy) treatment on ACAG. To account for multiple testing, significance thresholds were adjusted via Bonferroni correction: $P < 0.004 (0.05/12)$ for Cox regressions and $P < 0.005 (0.05/9)$ for subgroup analyses. All statistical analyses were performed using R software (version 4.2.2, R Foundation for Statistical Computing, Vienna, Austria) and MSTATA software (https://www.mstata.com/). The code used is available in the supplementary material "R.zip".

## 3. Results

### 3.1. Characteristics of patients at baseline

The study encompassed a total of 2,876 patients. The mean age of the patients was 59 years, with 64.0% males and 65.5% whites. When compared to the survivor group, the mortality group exhibited higher values in age, heart rate, respiratory rate, RDW, white blood cell count, AG, total calcium, ACAG, INR, sodium ions, BUN, total bilirubin, and creatinine. Moreover, their MELD and SOFA scores were also elevated (28.0 vs. 18.0, 10.0 vs. 8.0). In terms of complications and treatments, the mortality group had a higher proportion of patients with sepsis (46.4% vs. 20.9%), AKI (73.0% vs. 54.6%), SBP (14.3% vs. 9.2%), ascites (59.2% vs. 47.1%), HRS (26.1% vs. 12.0%), HE (12.3% vs. 9.5%), CRRT (25.7% vs.

11.3%), and the use of ventilators (84.6% vs. 78.1%). However, no statistically significant differences were found between the two groups regarding gender, diabetes, and variceal bleeding (P ≥ 0.05). The baseline characteristics of patients in both groups are presented in **Table 1**.

### 3.2. Cox regression analysis of ACAG and mortality in cirrhosis patients

The Cox proportional hazards model demonstrated a consistent association between elevated ACAG and increased mortality risk across all assessed time periods. Specifically, compared to the normal group, elevated ACAG was linked to higher mortality rates at 30 days (HR = 2.62, 95% CI: 2.30–3.00, P < 0.001), 90 days (HR = 2.48, 95% CI: 2.18–2.81, P < 0.001), 180 days (HR = 2.49, 95% CI: 2.19–2.82, P < 0.001), and 365 days (HR = 2.50, 95% CI: 2.21–2.83, P < 0.001).

In multivariable analysis, Model 1 adjusted for age and gender, and the elevated ACAG group still exhibited significantly elevated mortality risks at each time point: 30 days (HR = 2.63, 95% CI: 2.30–3.01, P < 0.001), 90 days (HR = 2.48, 95% CI: 2.19–2.82, P < 0.001), 180 days (HR = 2.49, 95% CI: 2.20–2.83, P < 0.001), and 365 days (HR = 2.51, 95% CI: 2.21–2.84, P < 0.001).

Moreover, Model 2, which accounted for additional covariates, confirmed the independent prognostic value of ACAG. The hazard ratios for the elevated ACAG group were 1.77 (95% CI: 1.52–2.06, P < 0.001) at 30 days, 1.64 (95% CI: 1.42–1.89, P < 0.001) at 90 days, 1.65 (95% CI: 1.43–1.90, P < 0.001) at 180 days, and 1.67 (95% CI: 1.45–1.92, P < 0.001) at 365 days.

These findings underscore the robust relationship between elevated ACAG levels and mortality risk in cirrhosis patients across different follow-up intervals, highlighting its potential as a significant prognostic indicator. For more details, refer to **Table 2**.

To bolster confidence in our findings, we performed a sensitivity analysis using the optimal cutoff value from the 30-day mortality ROC curve. Specifically, we analyzed ACAG at a cutoff value of 18.375. After adjusting for all potential confounding covariates, all P-values across the assessed time points were P < 0.001, underscoring the stability and significance of our results. For further details, see S1 Table.

### 3.3. Analysis of Kaplan-Meier and ROC Curves

The K-M survival analysis revealed significant differences in mortality rates between patients with elevated ACAG levels and those with normal levels at 30 days (47.5% vs. 28.9%, P < 0.001), 90 days (51.0% vs. 26.0%, P < 0.001), 180 days (51.6% vs. 26.2%, P < 0.001), and 365 days (52.3% vs. 26.6%, P < 0.001). These results are shown in **Fig 2**.

Compared to AG or albumin alone, the ACAG demonstrated better predictive value for both short- and long-term mortality.

For 30-day, 90-day, 180-day, and 365-day mortality predictions, ACAG achieved AUCs of 0.671 (CI: 0.649–0.693), 0.663 (CI: 0.641–0.684), 0.664 (CI: 0.643–0.685), and 0.665 (CI: 0.643–0.686), respectively, consistently outperforming AG (0.651, 0.647, 0.649, 0.649) and albumin (0.541, 0.534, 0.534, 0.534).

The predictive performance was further enhanced when ACAG was combined with SOFA or MELD scores. The SOFA+ACAG model achieved AUCs of 0.704, 0.699, 0.701, and 0.700 for 30-day, 90-day, 180-day, and 365-day mortality, respectively. Similarly, the MELD+ACAG combination demonstrated robust performance, with AUCs of 0.710, 0.709, 0.711, and 0.711, respectively. More details are provided in **Table 3** and **Fig 3**.

The decision curve analysis (DCA) for SOFA versus SOFA+ACAG and MELD versus MELD+ACAG across four follow-up periods (30-day, 90-day, 180-day, and 365-day) is presented in **Supplementary Figures 1** and 2 (S1 Fig and S2 Fig). At all time points, the curves for SOFA+ACAG and MELD+ACAG showed superior net benefit compared to SOFA and MELD alone, respectively. This indicates that incorporating ACAG into the MELD and SOFA scoring systems enhances mortality risk stratification and clinical utility.

**Table 1. Patient demographics and baseline characteristics.**

| Variables | All patients(n = 2826) | Survivors (n = 1956) | Nonsurvivors (n = 870) | p-value |
|---|---|---|---|---|
| **Demographic** | | | | |
| Age(year) | 59 (52, 67) | 59 (51, 66) | 61 (53, 68) | <0.001 |
| Gender, n (%) | | | | 0.386 |
| Male | 1810 (64.0%) | 1263 (64.6%) | 547 (62.9%) | |
| Female | 1016 (36.0%) | 693 (35.4%) | 323 (37.1%) | |
| Race, n (%) | | | | 0.007 |
| White | 1,852 (65.5%) | 1312 (67.1%) | 540 (62.1%) | |
| Other | 735 (26.0%) | 475 (24.3%) | 260 (29.9%) | |
| Black | 239 (8.5%) | 169 (8.6%) | 70 (8.0%) | |
| **Vital Signs** | | | | |
| Heart rate (beats/min) | 88 (77, 100) | 87 (76, 98) | 91 (79, 103) | <0.001 |
| Systolic blood pressure (mmHg) | 111 (101, 118) | 113 (102, 121) | 108 (98, 116) | <0.001 |
| Diastolic Blood Pressure (mmHg) | 63 (56, 68) | 63 (57, 69) | 60 (53, 65) | <0.001 |
| Respiratory rate (beats/min) | 18.0 (16.0, 21.0) | 18.0 (16.0, 20.3) | 19.0 (17.0, 23.0) | <0.001 |
| Temperature (°C) | 36.78 (36.59, 37.04) | 36.82(36.64,37.07) | 36.72 (36.49, 36.91) | <0.001 |
| **Laboratory Indicators** | | | | |
| hemoglobin(g/dl) | 9.16(8.03, 10.51) | 9.30 (8.15, 10.63) | 8.94 (7.83, 10.25) | <0.001 |
| platelet(10^9/L) | 93 (62, 145) | 94 (64, 145) | 89 (58, 144) | 0.008 |
| RDW (%) | 17.08 (15.49, 19.15) | 16.77(15.30,18.78) | 17.85 (16.00, 20.05) | <0.001 |
| RBC (10^9/L) | 2.95 (2.55, 3.39) | 3.01 (2.61, 3.43) | 2.82 (2.46, 3.27) | <0.001 |
| WBC (10^9/L) | 10 (6, 14) | 9 (6, 13) | 11 (7, 17) | <0.001 |
| Albumin(mg/dl) | 2.95 (2.53, 3.40) | 3.00 (2.60, 3.40) | 2.90 (2.45, 3.40) | <0.001 |
| Anion gap (m Eq/l) | 14.7 (12.0, 18.0) | 14.0 (11.7, 16.8) | 16.5 (13.0, 20.0) | <0.001 |
| TCa (mg/dl) | 8.30 (7.80, 8.85) | 8.27 (7.80, 8.80) | 8.33 (7.80, 8.95) | 0.032 |
| glucose (mg/dl) | 131 (106, 172) | 133 (108, 181) | 125 (100, 157) | <0.001 |
| Potassium (m Eq/l) | 4.15 (3.80, 4.63) | 4.12 (3.80, 4.55) | 4.23 (3.80, 4.80) | <0.001 |
| Sodium (m Eq/l) | 137 (133, 141) | 137 (134, 141) | 137 (132, 140) | <0.001 |
| INR | 1.70 (1.43, 2.15) | 1.63 (1.40, 2.00) | 2.00 (1.60, 2.50) | <0.001 |
| ALT(IU/L) | 35 (20, 90) | 35 (20, 104) | 36 (21, 74) | 0.413 |
| AST(IU/L) | 74 (41, 191) | 70 (40, 198) | 79 (42, 175) | 0.321 |
| Total bilirubin(mg/dl) | 3 (1, 7) | 3 (1, 6) | 4 (2, 12) | <0.001 |
| creatinine(mg/dl) | 1.30 (0.85, 2.25) | 1.17 (0.80, 1.90) | 1.80 (1.06, 3.10) | <0.001 |
| BUN mg/dl | 28 (17, 47) | 25 (16, 42) | 36 (21, 60) | <0.001 |
| SOFA | 8.0 (6.0, 11.0) | 8.0 (5.0, 10.0) | 10.0 (8.0, 14.0) | <0.001 |
| MELD | 21 (13, 28) | 18 (12, 25) | 26 (18, 33) | <0.001 |
| ACAG (m Eq/l) | 18.0 (15.8, 21.3) | 17.5 (15.3, 20.0) | 20.3 (17.1, 23.5) | <0.001 |
| **Comorbidities** | | | | |
| Sepsis, (n %) | 813 (28.8%) | 409 (20.9%) | 404 (46.4%) | <0.001 |
| AKI, n (%) | 1,703 (60.3%) | 1,068 (54.6%) | 635 (73.0%) | <0.001 |
| Variceal Bleeding, n (%) | 168 (5.9%) | 118 (6.0%) | 50 (5.7%) | 0.767 |
| SBP, n (%) | 304 (10.8%) | 180 (9.2%) | 124 (14.3%) | <0.001 |
| Ascites, n (%) | 1,437 (50.8%) | 922 (47.1%) | 515 (59.2%) | <0.001 |
| HRS, n (%) | 457 (16.2%) | 45 (12.0%) | 35 (26.1%) | <0.001 |
| HE, n (%) | 293 (10.4%) | 186 (9.5%) | 107 (12.3%) | 0.025 |
| Diabetes, n (%) | 824 (29.2%) | 561 (28.7%) | 263 (30.2%) | 0.403 |
| Heart Failure, n (%) | 500 (17.7%) | 325 (16.6%) | 175 (20.1%) | 0.024 |

*(Continued)*

**Table 1.** (Continued)

| Variables | All patients(n = 2826) | Survivors (n = 1956) | Nonsurvivors (n = 870) | p-value |
|---|---|---|---|---|
| **Treatment** | | | | |
| Ventilator, n (%) | 2,263 (80.1%) | 1,527 (78.1%) | 736 (84.6%) | <0.001 |
| CRRT, n (%) | 445 (15.7%) | 221 (11.3%) | 224 (25.7%) | <0.001 |

T Ca: Total calcium; BUN: Blood Urea Nitrogen; AST: aspartate aminotransferase; ALT: alanine aminotransferase; RDW: Red Cell Distribution Width; RBC: Red Blood Cell; WBC: white blood cell; INR: international normalized ratio; MELD: Model for End-Stage Liver Disease; CRRT: continuous renal replacement therapy; AKI: acute kidney injury; ACAG: Albumin Corrected Anion Gap; HE: Hepatic Encephalopathy; SOFA, Sepsis-related Organ Failure Assessment score. SBP: Spontaneous bacterial peritonitis.

**Table 2.** Association between ACAG and mortality in patients with cirrhosis.

| Outcome | Unadjusted HR (95%CI) | Model1 HR (95%CI) | Model2 HR (95%CI) |
|---|---|---|---|
| **30-d** | | | |
| ACAG≤20 | 1 | 1 | 1 |
| ACAG>20 | 2.62(2.30, 3.00) | 2.63(2.30, 3.01) | 1.77(1.52, 2.06) |
| P | <0.001 | <0.001 | <0.001 |
| **90-d** | | | |
| ACAG≤20 | 1 | 1 | 1 |
| ACAG>20 | 2.48(2.18, 2.81) | 2.48(2.19, 2.82) | 1.64(1.42, 1.89) |
| P | <0.001 | <0.001 | <0.001 |
| **180-d** | | | |
| ACAG≤20 | 1 | 1 | 1 |
| ACAG>20 | 2.49(2.19, 2.82) | 2.49(2.20, 2.83) | 1.65(1.43, 1.90) |
| P | <0.001 | <0.001 | <0.001 |
| **365-d** | | | |
| ACAG≤20 | 1 | 1 | 1 |
| ACAG>20 | 2.50(2.21, 2.83) | 2.51(2.21, 2.84) | 1.67(1.45, 1.92) |
| P | <0.001 | <0.001 | <0.001 |

Model 1: Adjusted age and Gender.

Model 2: Model 1 + Temperature, Platelet, WBC, Sodium, INR, BUN, Sepsis, Variceal bleeding, SBP, Ascites, HRS, HE, AKI, and CRRT.

### 3.4. Restricted Cubic Spline Analysis of ACAG in Relation to Cirrhosis Prognosis

In our study, we applied RCS analysis to identify the nonlinear relationships between ACAG and the prognosis of cirrhosis at different time points. The results at specific time points are as follows: 30 days (P<0.001, P-Nonlinear=0.622), 90 days (P<0.001, P-Nonlinear=0.429), 180 days (P<0.001, P-Nonlinear=0.434), and 365 days (P<0.001, P-Nonlinear=0.416). These findings suggest that an increase in ACAG is associated with a linear increase in mortality rates across all evaluated time points. For more detailed results, please refer to **Fig 4**.

### 3.5. Subgroup analyses of ACAG in Cirrhosis patients

In our subgroup analyses, we evaluated the influence of various factors, including age, sex, race, sepsis, AKI, variceal bleeding, SBP, HE, HRS and CRRT, on the outcomes. We observed no significant interaction between these factors and ACAG, indicating the robustness of the association between ACAG and cirrhosis prognosis (all interaction P-values >0.005). These detailed findings are presented in **Fig 5**.

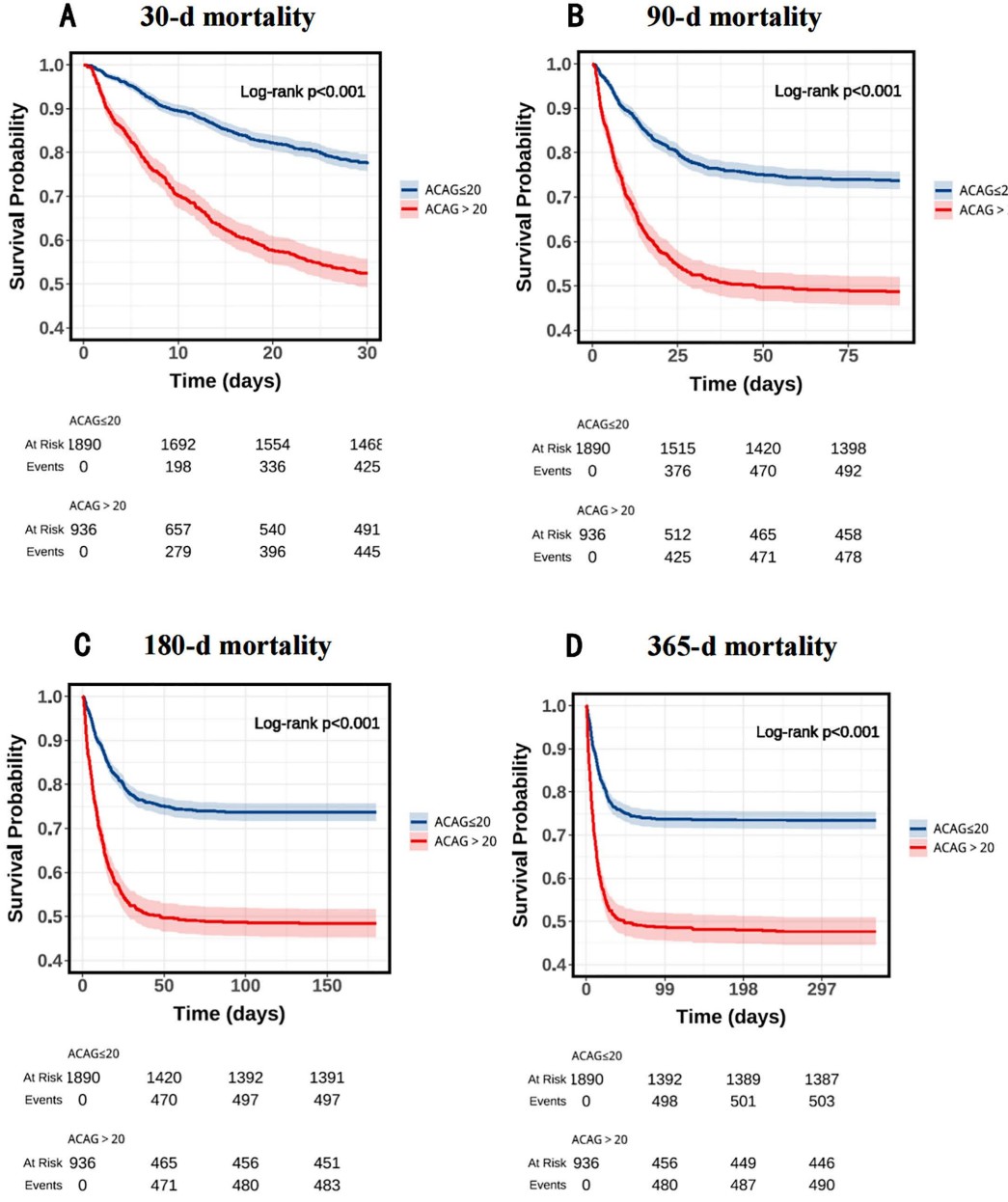

**Fig 2. K-M survival analysis curves for all-cause mortality in patients with cirrhosis at 30-d (A), 90-d (B),180-d (C) and 365-d (D) of hospital admission.**

## 4. Discussion

Liver cirrhosis is associated with a poor prognosis, marked by high rates of morbidity and mortality [24]. In the ICU setting, patients with liver cirrhosis face an even more dismal outlook due to the severity of their condition and the frequent coexistence of multiple comorbidities [25]. Accurate prognosis assessment is thus critical for informing clinical decisions and enhancing patient outcomes. Currently, ICUs utilize multiple prognostic scoring systems, such as the SOFA score and the MELD score, to evaluate disease severity. However, these systems are often complex and time-consuming to calculate,

**Table 3. Information of ROC curves in Fig 3.**

| Variable | AUC (%) | 95%CI (%) | Threshold | Sensitivity | Specificity |
|---|---|---|---|---|---|
| **30-d** | | | | | |
| **ACAG** | 0.671 | 0.649 - 0.693 | 18.375 | 0.571 | 0.700 |
| **AG** | 0.651 | 0.628 - 0.673 | 17.200 | 0.461 | 0.783 |
| **Albumin** | 0.541 | 0.517 - 0.564 | 2.600 | 0.368 | 0.724 |
| **SOFA** | 0.680 | 0.658 - 0.702 | 9.000 | 0.670 | 0.581 |
| **MELD** | 0.692 | 0.671 - 0.713 | 24.000 | 0.585 | 0.585 |
| **SOFA+ACAG** | 0.704 | 0.683 - 0.725 | | 0.278 | 0.928 |
| **MELD+ACAG** | 0.710 | 0.690 - 0.731 | | 0.271 | 0.923 |
| **90-d** | | | | | |
| **ACAG** | 0.663 | 0.641 - 0.684 | 18.375 | 0.551 | 0.705 |
| **AG** | 0.647 | 0.625 - 0.669 | 15.500 | 0.595 | 0.645 |
| **Albumin** | 0.534 | 0.511 - 0.557 | 2.600 | 0.362 | 0.726 |
| **SOFA** | 0.676 | 0.655 - 0.697 | 9.000 | 0.661 | 0.591 |
| **MELD** | 0.695 | 0.675 - 0.716 | 24.000 | 0.561 | 0.545 |
| **SOFA+ACAG** | 0.699 | 0.679 - 0.720 | | 0.323 | 0.904 |
| **MELD+ACAG** | 0.709 | 0.689 - 0.730 | | 0.328 | 0.903 |
| **180-d** | | | | | |
| **ACAG** | 0.664 | 0.643 - 0.685 | 18.375 | 0.552 | 0.706 |
| **AG** | 0.649 | 0.627 - 0.670 | 15.500 | 0.596 | 0.646 |
| **Albumin** | 0.534 | 0.511 - 0.557 | 2.600 | 0.361 | 0.726 |
| **SOFA** | 0.677 | 0.656 - 0.698 | 9.000 | 0.661 | 0.592 |
| **MELD** | 0.696 | 0.676 - 0.717 | 24.000 | 0.562 | 0.750 |
| **SOFA+ACAG** | 0.701 | 0.680 - 0.721 | | 0.329 | 0.904 |
| **MELD+ACAG** | 0.711 | 0.691 - 0.731 | | 0.331 | 0.900 |
| **365-d** | | | | | |
| **ACAG** | 0.665 | 0.643 - 0.686 | 19.455 | 0.470 | 0.789 |
| **AG** | 0.649 | 0.627 - 0.671 | 15.670 | 0.573 | 0.672 |
| **Albumin** | 0.534 | 0.511 - 0.557 | 2.600 | 0.360 | 0.726 |
| **SOFA** | 0.676 | 0.655 - 0.697 | 9.000 | 0.660 | 0.592 |
| **MELD** | 0.696 | 0.675 - 0.716 | 24.000 | 0.559 | 0.750 |
| **SOFA+ACAG** | 0.700 | 0.680 - 0.721 | | 0.333 | 0.901 |
| **MELD+ACAG** | 0.711 | 0.691 - 0.731 | | 0.338 | 0.897 |

ROC, receiver operating characteristic; AUC, area under the curve; CI, confidence interval; AG, Anion Gap; ACAG, Albumin Corrected Anion Gap; MELD, Model for End-Stage Liver Disease; SOFA, Sepsis-related Organ Failure Assessment score.

limiting their rapid deployment in clinical practice. While recent research has focused on developing simpler indicators like the international normalized ratio-to-albumin ratio [26], lactate-to-albumin ratio [27], and neutrophil-to-lymphocyte ratio [28] to facilitate rapid prognosis assessment in patients with liver cirrhosis, there remains a critical need for even more straightforward and accessible indicators to guide clinicians in the timely management of critically ill patients.

In this study, univariate and multivariate analyses were performed on critically ill patients with liver cirrhosis, revealing that the ACAG serves as an independent predictor of both short-term and long-term all-cause mortality in severe liver cirrhosis. Patients were stratified into two groups based on ACAG levels: normal AG and high AG. K-M survival analysis demonstrated significantly worse short-term and long-term prognoses in the high ACAG group. ACAG levels exhibited

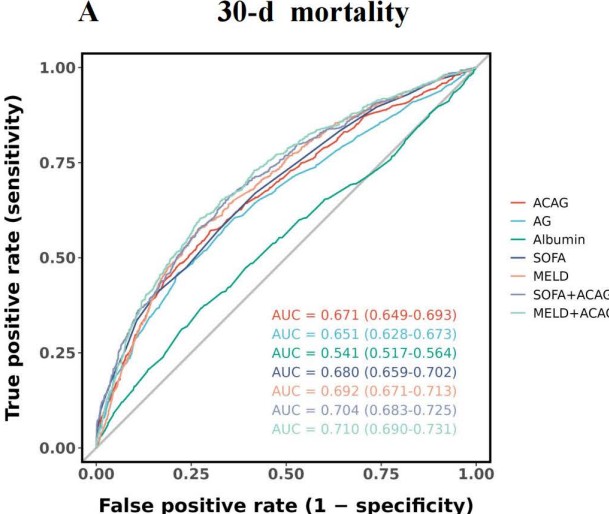

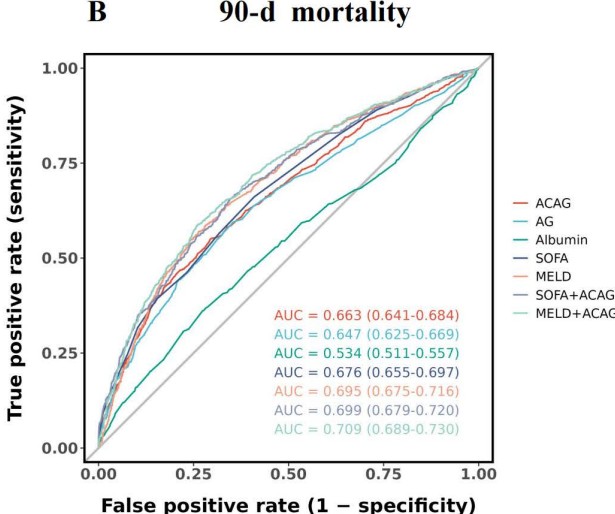

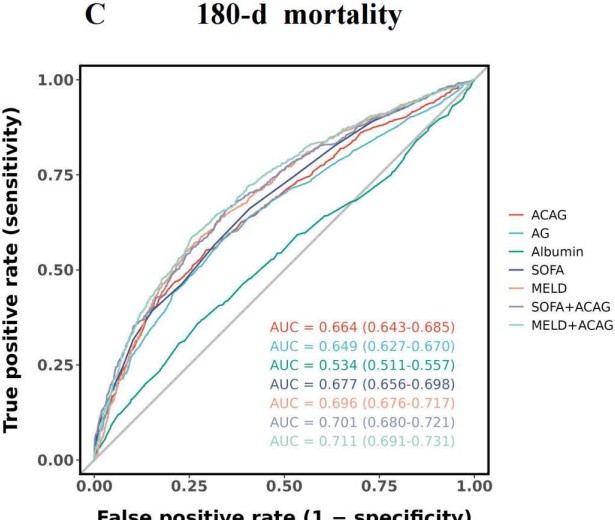

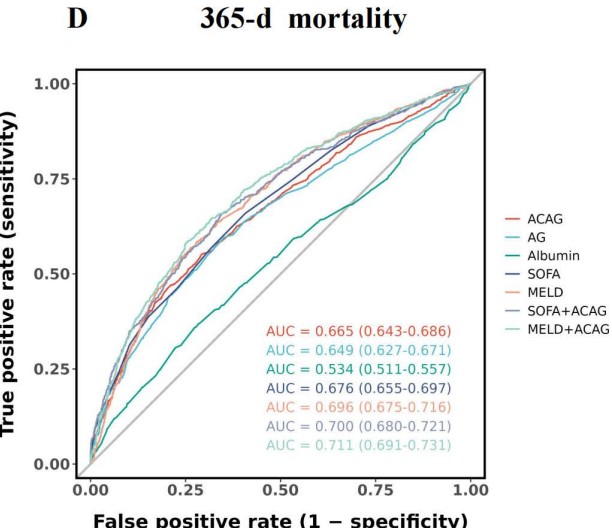

**Fig 3. ROC curves for predicting all-cause mortality in patients with cirrhosis at 30-d (A), 90-d (B),180-d (C) and 365-d (D) of hospital admission.**

moderate discriminative ability in predicting mortality at multiple time points in cirrhotic patients. However, the current evidence suggests that ACAG may serve as a potential supplementary tool rather than outperform well-validated scoring systems such as SOFA or MELD. Additionally, RCS analysis identified a linear relationship between ACAG levels and cirrhosis prognosis, indicating that elevated ACAG is associated with poorer outcomes. Subgroup analysis further confirmed the absence of significant interactions between ACAG and other prognostic factors in severe liver cirrhosis, underscoring the reliability of ACAG as a robust prognostic marker.

Multiple studies have established a robust correlation between the ACAG and disease prognosis across various conditions. For instance, a study involving 344 patients with acute pancreatitis by Li et al [29]. revealed that individuals with ACAG levels exceeding 19 exhibited a significantly higher in-hospital mortality rate (HR: 3.46), underscoring the

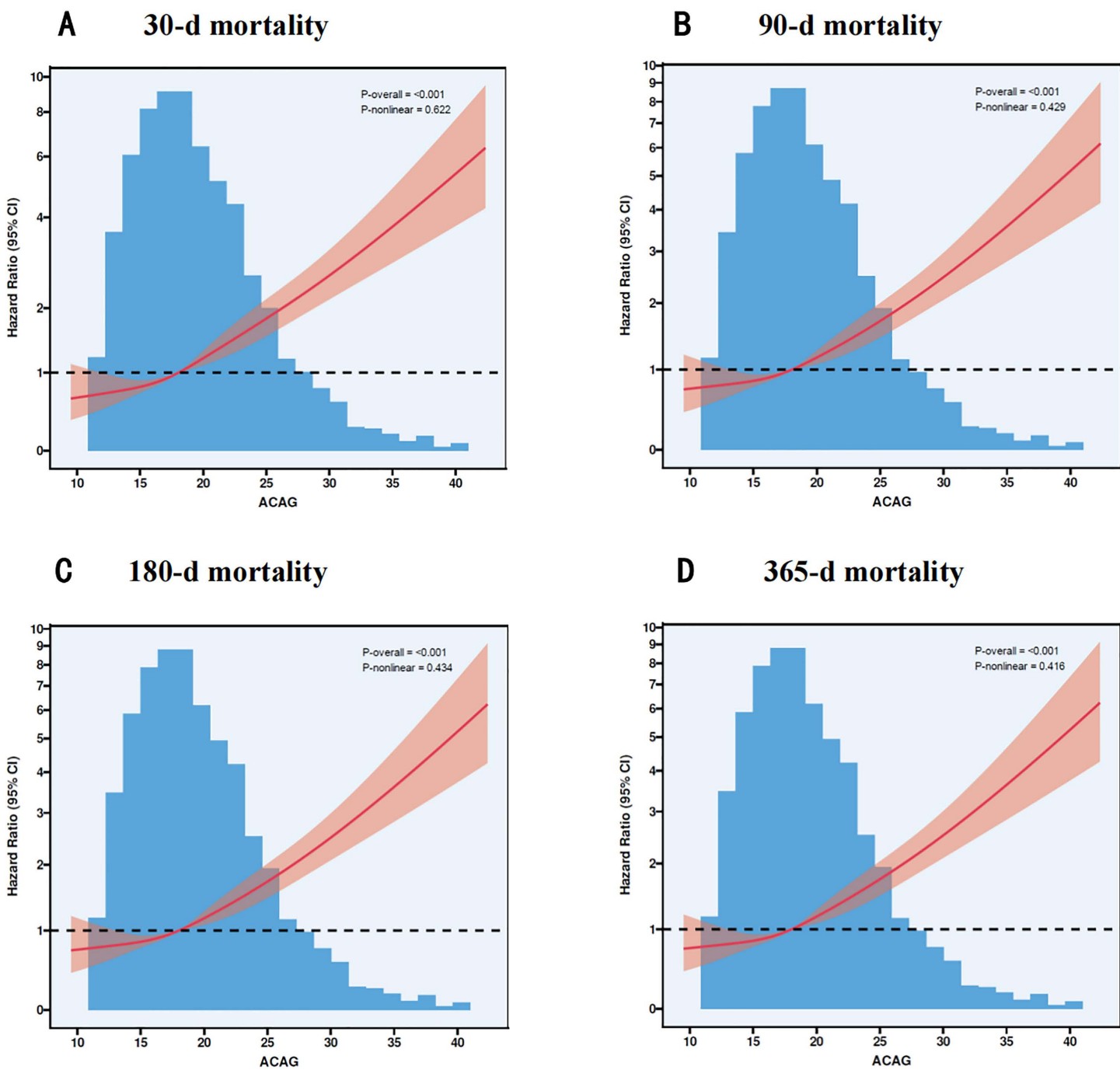

**Fig 4. Association between ACAG and Survival with the RCS function at 30-d (A), 90-d (B), 180-d (C) and 365-d (D) after admission.**

independent association between elevated ACAG and increased mortality risk in this population. Similarly, a retrospective analysis of 1,561 asthma patients by Wang et al. identified elevated ACAG as an independent predictor of 30-day outcomes in critically ill asthma patients (HR: 1.07), with ACAG demonstrating superior predictive accuracy compared to the traditional anion gap [22]. Additionally, a retrospective study [30] found that elevated ACAG was a significant risk factor

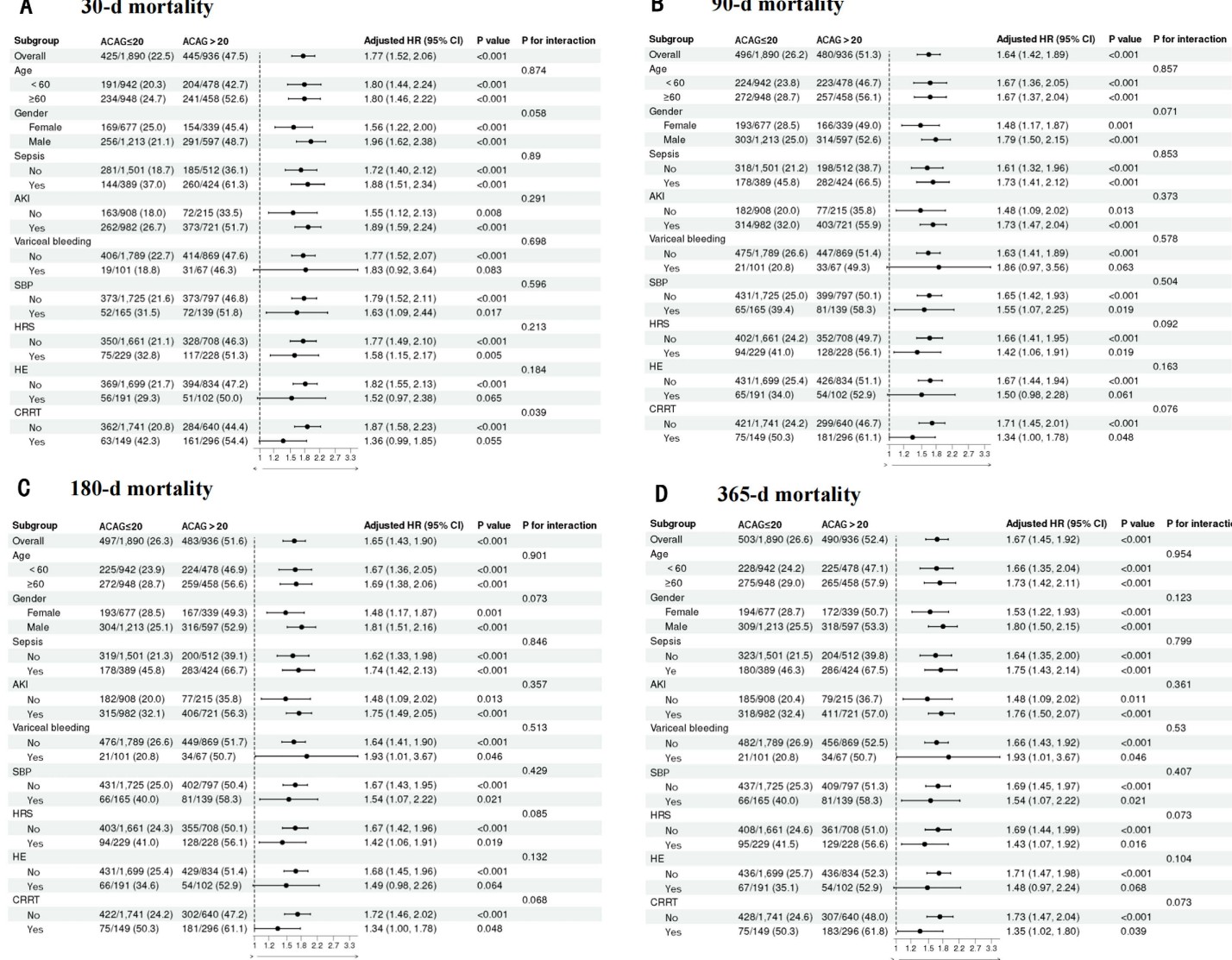

**Fig 5. Forest plots of subgroup analysis of the relationship between all-cause mortality and ACAG in patients with cirrhosis at admission 30-d (A), 90-d (B), 180-d (C), and 365-d (D).**

for 30-day all-cause mortality in critically ill patients with acute myocardial infarction (HR: 1.75). Comparable associations have been reported in other conditions, including sepsis [31], cardiogenic shock [23], and acute kidney injury [32]. Collectively, these findings affirm ACAG as a reliable biochemical marker and provide a robust clinical foundation for our research.

The AG is a fundamental parameter for evaluating acid-base disturbances, particularly in diagnosing metabolic acidosis in cirrhosis [33]. Progressive hepatic dysfunction in these patients leads to metabolic derangements, including: (1) impaired lactate metabolism—causing systemic accumulation, Type A lactic acidosis (secondary to tissue hypoxia and hemodynamic instability) [34,35], and Type B lactic acidosis (due to reduced hepatic perfusion and clearance) [36,37], often manifesting as mixed lactic acidosis with elevated AG; (2) impaired ketone regulation, which may precipitate

ketoacidosis, especially in diabetics or insulin-deficient individuals [38,39]]; and (3) renal insufficiency, a common complication in advanced cirrhosis that exacerbates acidosis [40]. These mechanisms underscore the importance of AG monitoring for early detection and intervention [41]. Additionally, hypoalbuminemia—a hallmark of cirrhosis—significantly impacts AG as albumin is a major unmeasured anion [12,42]. Conventional AG may underestimate acid-base disturbances in these patients, necessitating albumin-corrected AG (ACAG) for accurate assessment [43].

The SOFA score, despite its widespread use in intensive care, has limited relevance to cirrhotic patients due to their unique pathophysiology [44]. Similarly, the MELD score, though it is the current standard for prognosis, only considers a few biochemical parameters (INR, creatinine, and bilirubin), so it cannot account for critical complications like hepatic encephalopathy, spontaneous bacterial peritonitis, and variceal hemorrhage [45,46]]. Recent modifications, such as incorporating serum sodium (MELD-Na) or additional variables like age and albumin (MELD 3.0), have shown improved prognostic performance, underscoring the need for score optimization [47,48].

The ACAG introduced in this study can mitigate these limitations to some extent. As a robust indicator of acid-base disturbances, ACAG elevation correlates strongly with metabolic acidosis severity, hepatic impairment, and adverse clinical outcomes. By incorporating albumin alongside anion gap measurements, ACAG provides a more comprehensive representation of patient pathophysiology. Importantly, our data show that ACAG enhances mortality risk prediction when combined with either MELD or SOFA scores, offering a valuable advancement in cirrhosis prognostic modeling.

Clinically, ACAG offers several distinct advantages. It enables rapid risk stratification of critically ill cirrhotic patients, facilitating timely clinical decision-making. Elevated values predict adverse outcomes and warrant aggressive intervention, while lower values may support therapeutic de-escalation. This risk stratification promotes resource optimization and personalized treatment. The practical implementation of ACAG is further enhanced by its reliance on routinely available laboratory measurements (albumin and anion gap), eliminating the need for additional testing while maintaining cost – effectiveness and clinical accessibility.

Our research is constrained by several notable limitations. Firstly, the investigation was conducted within a single-center framework. Although we integrated an extensive sample of real-world data, the applicability of our findings may be limited across varied geographic regions and heterogeneous populations. Secondly, the study did not evaluate the severity grading or the underlying etiology of the disease, factors which could potentially narrow the scope of our results' interpretability. Moreover, we were unable to determine the definitive causes of mortality for all patients, as death in critically ill individuals is frequently attributable to a confluence of multiple factors. Lastly, the study population exclusively comprised patients admitted to the ICU, a characteristic that may introduce selection bias and limit the generalizability of our conclusions.

## 5. Conclusion

This study demonstrates that ACAG levels in cirrhotic patients are significantly associated with both short- and long-term clinical outcomes. Further validation through large-scale, multicenter prospective cohort studies is warranted to confirm these findings and evaluate the broader applicability of ACAG across heterogeneous patient populations.

## Supporting information

**S1 Table.**   Results of the sensitivity analysis.
(DOCX)

**S1 File.  R code used for statistical analyses.**
(RAR)

**S2 File.  Database.** Source database used for the analyses in this study.
(CSV)

**S1 Fig.** DCA curves for SOFA versus SOFA+ACAG in patients with cirrhosis at 30-d (A), 90-d (B), 180-d (C), and 365-d (D) of hospital admission.
(TIF)

**S2 Fig. DCA curves for MELD versus MELD+ACAG in patients with cirrhosis at 30-d (A), 90-d (B), 180-d (C), and 365-d (D) of hospital admission.**
(TIF)

## Author contributions

**Data curation:** Ce Xu, Guangdong Wang, Lihong Lv, Mengyuan Chen.

**Formal analysis:** Ce Xu, Guangdong Wang, Min Zhang, Xingyi Yang.

**Resources:** Ce Xu.

**Software:** Min Zhang, Mengyuan Chen.

**Writing – original draft:** Ce Xu, Xingyi Yang.

**Writing – review & editing:** Xingyi Yang.

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
