## [Decision Letter · Decision Letter 0]

21 May 2025

PONE-D-25-10489Albumin-Corrected Anion Gap as a Predictive Marker for Mortality in Critically Ill Cirrhosis Patients: An Analysis Based on the MIMIC-IV DatabasePLOS ONE

Dear Dr. yang,

Thank you for submitting your manuscript to PLOS ONE. After careful consideration, we feel that it has merit but does not fully meet PLOS ONE’s publication criteria as it currently stands. Therefore, we invite you to submit a revised version of the manuscript that addresses the points raised during the review process.

<h3 data-end="973" data-start="931">**Required Revisions for Acceptance:** </h3>

**Clarification of Cirrhosis Diagnosis (ICD Codes)**Please explicitly list the ICD-9/10 codes used to define cirrhosis in the Methods section (currently not specified).**Handling of Missing Data**Expand on your current explanation of missing data. You mention Random Forest imputation for ≤20% missingness—please justify this choice and specify which variables were imputed.

**Normality and Data Presentation**Clarify whether tests for normality (e.g., Shapiro-Wilk) were performed. If data were not normally distributed, justify the use of mean ± SD over median (IQR) for continuous variables.

**Multivariate Model Construction**Clearly describe the variable selection process for Model 2. State whether stepwise selection, expert judgment, or univariate p-values were used. Also confirm whether multicollinearity (e.g., using VIF) was assessed.

**Code and Dataset Availability**Upload all relevant R code (particularly for ROC analysis, multivariate models, RCS plots, and subgroup analysis) and a de-identified dataset (in CSV format) as supplementary material. These are needed for transparency and reproducibility.

**Subgroup Analysis: Interaction Testing**Clarify whether formal tests for interaction were conducted (i.e., inclusion of interaction terms in Cox models). If not, please either perform and report p-values for interaction or state explicitly why these were not included.

**Discussion and Interpretation of AUC Values**Temper the language in your conclusions. The ROC AUC for ACAG (0.66–0.67) is modest. Please clarify that ACAG may complement—but not outperform—established tools like SOFA or MELD.

**Revision of Discussion Section**Improve contextualization of your findings with prior literature on cirrhosis-related prognostic tools. Streamline redundant pathophysiology and highlight clinical relevance of ACAG in resource-limited settings.

**Revision of Conclusions**Please soften your conclusions to reflect limitations inherent in a retrospective single-center database analysis (e.g., causal inference, external generalizability).

**Validation of X-tile Cutoff**

You report an ACAG cutoff of >20 derived via X-tile. Please discuss its clinical applicability and consider a sensitivity analysis (e.g., evaluating model performance across other potential thresholds).

<h3 data-end="3431" data-start="3370">**Recommended Revisions (Not Mandatory for Acceptance):** </h3>

**Comparative Prognostic Value**Provide clearer insights into how ACAG might be used alongside MELD and SOFA. A decision curve analysis or net reclassification index could add depth.

**Discussion Clarity**The discussion on acid-base physiology, lactic/ketoacidosis, and albumin’s role is scientifically accurate but overly repetitive. Condense and focus on clinical applicability.

**Minor Language Editing**The manuscript would benefit from a thorough English language edit to address minor grammar, punctuation, and typographical issues.

We look forward to receiving your revised manuscript.

Kind regards,

Manasa Kandula

Academic Editor

PLOS ONE

Journal requirements: 1. When submitting your revision, we need you to address these additional requirements. Please ensure that your manuscript meets PLOS ONE's style requirements, including those for file naming. The PLOS ONE style templates can be found at https://journals.plos.org/plosone/s/file?id=wjVg/PLOSOne_formatting_sample_main_body.pdf and https://journals.plos.org/plosone/s/file?id=ba62/PLOSOne_formatting_sample_title_authors_affiliations.pdf 2. We note that there is identifying data in the Supporting Information file “AGAG”. Due to the inclusion of these potentially identifying data, we have removed this file from your file inventory. Prior to sharing human research participant data, authors should consult with an ethics committee to ensure data are shared in accordance with participant consent and all applicable local laws. Data sharing should never compromise participant privacy. It is therefore not appropriate to publicly share personally identifiable data on human research participants. The following are examples of data that should not be shared: -Name, initials, physical address-Ages more specific than whole numbers-Internet protocol (IP) address-Specific dates (birth dates, death dates, examination dates, etc.)-Contact information such as phone number or email address-Location data-ID numbers that seem specific (long numbers, include initials, titled “Hospital ID”) rather than random (small numbers in numerical order) Data that are not directly identifying may also be inappropriate to share, as in combination they can become identifying. For example, data collected from a small group of participants, vulnerable populations, or private groups should not be shared if they involve indirect identifiers (such as sex, ethnicity, location, etc.) that may risk the identification of study participants. Additional guidance on preparing raw data for publication can be found in our Data Policy (https://journals.plos.org/plosone/s/data-availability#loc-human-research-participant-data-and-other-sensitive-data) and in the following article: http://www.bmj.com/content/340/bmj.c181.long. Please remove or anonymize all personal information (<specific identifying information in file to be removed>), ensure that the data shared are in accordance with participant consent, and re-upload a fully anonymized data set. Please note that spreadsheet columns with personal information must be removed and not hidden as all hidden columns will appear in the published file. 3. Please include captions for your Supporting Information files at the end of your manuscript, and update any in-text citations to match accordingly. Please see our Supporting Information guidelines for more information: http://journals.plos.org/plosone/s/supporting-information.

Reviewers' comments:

Reviewer's Responses to Questions

**Comments to the Author**

1. Is the manuscript technically sound, and do the data support the conclusions?

Reviewer #1: Yes

Reviewer #2: Yes

Reviewer #3: Yes

2. Has the statistical analysis been performed appropriately and rigorously? 

Reviewer #1: Yes

Reviewer #2: Yes

Reviewer #3: Yes

3. Have the authors made all data underlying the findings in their manuscript fully available?

Reviewer #1: No

Reviewer #2: Yes

Reviewer #3: Yes

4. Is the manuscript presented in an intelligible fashion and written in standard English?

Reviewer #1: Yes

Reviewer #2: Yes

Reviewer #3: Yes

5. Review Comments to the Author

**Reviewer #1:**  This study "Albumin-Corrected Anion Gap as a Predictive Marker for Mortality in Critically Ill Cirrhosis Patients" explores a key and clinically relevant topic, investigating Albumin-Corrected Anion Gap (ACAG) as a predictive marker for mortality in critically ill cirrhotic patients. While this study utilizes a robust dataset (MIMIC-IV), several critical areas must be addressed before the manuscript can be considered for publication.

1) The diagnosis was based on the relevant International Classification of Diseases (ICD) codes. Which ICD codes were used to determine the diagnosis of cirrhosis?

2) As a retrospective study, missing values are inevitable. Could you please elaborate on how these missing values were handled in the method section?

3) In large databases like MIMIC-IV, data are usually not normally distributed. What are the rationales for the authors presenting the data as mean and standard deviation? Did the authors check the normality of the data?

4) Please explain how the variables were selected for multivariate analysis.

5) Did the study account for the potential impact of medications, such as diuretics or corticosteroids, on ACAG levels in critically ill patients?

6) The authors stated that they developed ROC curves using five parameters—ACAG, AG, ALB, SOFA score, and MELD score—to evaluate their predictive ability. However, the ROC curves did not perform well. Could the authors explain why these parameters did not yield better predictive performance?

7) Please provide the data in CSV files and the R code for performing the Restricted Cubic Spline Analysis of ACAG concerning cirrhosis prognosis.

8) Line 255: The authors stated that 'In our subgroup analyses, we evaluated the influence of various factors, including age, sex, race, sepsis, AKI, variceal bleeding, SBP, HE, HRS, and CRRT, on the outcomes.' Did the authors adjust for any stratified variables? If so, please provide the data and the R code used for the subgroup analysis.

9) Please provide the dataset used in the analysis (in CSV or Excel format) along with the full R code used to generate all tables and figures. The authors can upload these as supplementary materials during the revision process, including both the final dataset and the R scripts.

10) Kindly revise the discussion section to highlight the current study's findings while placing them in the context of existing literature on related diseases i.e. cirrhosis. The updated discussion should provide a critical analysis of how the results compare to or differ from previous research, focusing on potential mechanisms and clinical implications and suggesting areas for future research.

11) Please revise the conclusions to present a more cautious interpretation of the study findings, considering the study design limitations and/or sample size.

**Reviewer #2:**  The study addresses a clinically relevant and timely question regarding the prognostic value of albumin-corrected anion gap (ACAG) in critically ill cirrhotic patients. Utilizing a large-scale dataset (MIMIC-IV) enhances the credibility and reproducibility of your findings. Overall, the study is methodologically sound, clearly presented, and ethically compliant.

However, several minor revisions are necessary to enhance the manuscript before publication:

1.Please perform careful proofreading to correct minor grammatical and typographical errors throughout the manuscript.

2.Clearly justify or discuss the validation and robustness of the cutoff value (ACAG>20) derived by X-tile software. Consider conducting sensitivity analyses or further discuss its applicability.

3.Provide a more thorough comparative analysis of ACAG with established prognostic models (MELD, SOFA scores). Clearly articulate how ACAG adds clinical value or could practically complement existing prognostic tools.

4.Address the moderate ROC AUC values (~0.66) explicitly, discussing their implications for clinical decision-making and the utility of ACAG as a standalone predictor.

**Reviewer #3: ** Clarify Model Building Strategy:

Please describe the rationale and method for selecting variables in multivariate models (Model 2).

Were variables checked for multicollinearity (e.g., VIF)?

Interaction Terms in Subgroup Analysis:

While the subgroup analysis is comprehensive, please clarify whether formal interaction tests were performed. If not, please include interaction p-values or state why they were omitted.

Interpretation of AUC Values:

The ROC AUC for ACAG (0.66–0.67) is modest and slightly inferior to MELD/SOFA. Please temper the conclusion and discuss ACAG as a complement—not a replacement—for established scores.

Discussion Clarity:

The Discussion is well-referenced but includes redundant explanations (e.g., repeated pathophysiology of acid-base balance and hypoalbuminemia). Consider streamlining.

Add brief commentary on how ACAG could potentially be used in clinical triage or early risk stratification, particularly in settings with limited resources.

Limitations Section:

This is already strong. Consider explicitly stating that causality cannot be inferred due to the retrospective observational design.

6. PLOS authors have the option to publish the peer review history of their article (what does this mean? ). If published, this will include your full peer review and any attached files.

**Do you want your identity to be public for this peer review?** For information about this choice, including consent withdrawal, please see our Privacy Policy .

Reviewer #1: **Yes: ** Mohan Giri

Reviewer #2: No

Reviewer #3: No

---

## [Author Response · Author response to Decision Letter 1]

2 Jun 2025

All modifications have been made in the WORD versions of the Response to Reviewers, the Revised Manuscript with Track Changes, and the supplementary materials.

---

## [Editor Report · Decision Letter 1]

2 Sep 2025

Albumin-Corrected Anion Gap as a Predictive Marker for Mortality in Critically Ill Cirrhosis Patients: An Analysis Based on the MIMIC-IV Database

PONE-D-25-10489R1

Dear Dr. yang,

We’re pleased to inform you that your manuscript has been judged scientifically suitable for publication and will be formally accepted for publication once it meets all outstanding technical requirements.

Kind regards,

Tarek Samy Abdelaziz, MD,FRCP

Academic Editor

PLOS ONE
---

## [Editor Report · Acceptance letter]

PONE-D-25-10489R1

PLOS ONE

Dear Dr. Yang,

I'm pleased to inform you that your manuscript has been deemed suitable for publication in PLOS ONE. Congratulations! Your manuscript is now being handed over to our production team.

Kind regards,

on behalf of

Professor Tarek Samy Abdelaziz

Academic Editor

PLOS ONE